# Association between White Matter T2 Hyper-Intense Signals in Fetal Brain Magnetic Resonance Imaging and Neurodevelopment of Fetuses with Cytomegalovirus Infection

**DOI:** 10.3390/diagnostics14080797

**Published:** 2024-04-11

**Authors:** Galia Barkai, Eldad Katorza, Simon Lassman, Itachi Levinberg, Chen Hoffmann, Omer Bar-Yosef

**Affiliations:** 1Pediatric Infectious Disease Unit, Edmond and Lili Safra Children’s Hospital, Chaim Sheba Medical Center, Ramat Gan 52621, Israel; galia.barkai@sheba.health.gov.il; 2School of Medicine, Tel Aviv University, Tel Aviv 6139001, Israelomer.baryosef@sheba.health.gov.il (O.B.-Y.); 3Sheba BEYOND, Israel’s First Virtual Hospital, Ramat Can 52621, Israel; 4Gertner Institute of Epidemiology & Health Policy Research, Chaim Sheba Medical Center, Ramat Gan 52621, Israel; 5Antenatal Diagnostic Unit, Department of Obstetrics and Gynecology, Chaim Sheba Medical Center, Tel-Hashomer, Ramat Gan 52621, Israel; 6Arrow Program for Medical Research Education, Chaim Sheba Medical Center, Tel-Hashomer, Ramat Gan 52621, Israel; 7Wolfson Medical Center, Holon School of Medicine, Faculty of Medical and Health Sciences, Tel Aviv University, Tel Aviv 6139001, Israel; 8Section of Neuroradiology, Division of Diagnostic Imaging, Chaim Sheba Medical Center, Tel-Hashomer, Ramat Can 52621, Israel; 9Pediatric Neurology Unit, Department of Pediatrics, Sheba Medical Center, Ramat Gan 52621, Israel

**Keywords:** fetus, brain, magnetic resonance imaging, cytomegalovirus, white matter

## Abstract

An association between subtle changes in T2 white matter hyper-intense signals (WMHSs) detected in fetal brain magnetic resonance imaging (fbMRI) and congenital cytomegalovirus (CMV) infection has been established. The research aim of this study is to compare children with congenital CMV infection with neurodevelopment outcome and hearing deficit with and without WMHSs in a historic prospective case study cohort of 58 fbMRIs. Of these, in 37 cases, fbMRI was normal (normal group) and WMHSs were detected in 21 cases (WMHS group). The median infection week of the WMHS group was earlier than the normal fbMRI group (8 and 17 weeks of gestation, respectively). The proportion of infants treated with valganciclovir in the WMHS group was distinctly higher. Hearing impairment was not significantly different between the groups. VABS scores in all four domains were within normal range in both groups. The median score of the motor skills corrected for week of infection was better in the WMHS group. A multivariate analysis using the week of infection interaction variable of WMHS and valganciclovir treatment showed better motor score outcomes in the valganciclovir treatment group despite an earlier week of infection. WMHSs were not associated with neurodevelopmental outcome and hearing deficit. In our cohort, valganciclovir treatment may have a protective effect on fetuses with WMHSs by improving neurodevelopmental outcome.

## 1. Introduction

The herpesvirus human cytomegalovirus (CMV) is the most common intra-uterine infection in developed countries with an incidence of 0.5–1% of live births [1]. CMV infection is the most common non-genetic risk factor for congenital sensorineural hearing loss and neurodevelopmental sequelae [2,3,4,5,6], including mental retardation and epilepsy [7,8]. Other potential sequelae of pregnancies with maternal CMV infection include microcephaly, intra-uterine growth retardation, hepato-splenomegaly, elevated liver enzymes, thrombocytopenia, purpura and chorioretinitis, and/or optic atrophy in approximately 10% of symptomatic infants [3]. 

While only a minority of infants with congenital CMV exhibit CMV-specific symptoms at birth, 40–58% of those will develop permanent sequelae [2]. The majority of infected infants (85–90%) show no clinical signs at birth, i.e., asymptomatic CMV infection [2,9]. Asymptomatic infants are expected to develop normally, but 10–15% will develop late-onset sensory neural hearing loss [10]. 

Treatment with the antiviral valganciclovir is the standard treatment in symptomatic neonates [11] since studies have shown that antiviral treatment positively influenced hearing outcomes by reducing further worsening in hearing thresholds and in some cases even modestly improving long-term hearing outcomes [11].

Most medical centers advise continuation of pregnancy following diagnosis of fetal CMV infection, when prenatal neuroimaging is normal [12], increasing the importance of understanding the sequelae of neuroimaging conclusions. Previous studies have shown how certain etiologies are associated with T2 white matter hyper-intense signals (WMHSs) in fetal brain magnetic resonance imaging (fbMRI), including congenital CMV infection [13]. 

Theories of what it represents range from local gliosis to white matter edema [14,15]. These theories are supported by the histologic findings of human and animal fetuses which demonstrate innate and adaptive immune activation [16]. Three previous studies have found that fetuses with congenital CMV and subtle white matter MRI findings had a favorable outcome. However, none of the studies had a standardized neurodevelopment assessment [17,18,19,20]. 

Our research question is whether WMHSs on the fbMRI of fetuses with congenital CMV are associated with a worse neurodevelopmental outcome compared to normal fbMRI. 

## 2. Materials and Methods

### 2.1. Subject Demographics

This is a historic prospective case control study of women referred to our tertiary medical center for fbMRI due to CMV fetal infection between 2011 and 2015. All cases are singleton pregnancies with congenital CMV infection confirmed by either presence of CMV DNA in amniocentesis performed during pregnancy or by the presence of CMV DNA in neonatal saliva (screen), confirmed by a urine specimen (gold-standard) during the first 2 weeks of life. CMV DNA testing was performed using a real-time PCR assay in Sheba’s Central Virology Laboratory. 

Demographic and clinical data were collected from the electronic medical records of each mother and child and, as required, missing information was supplemented by a telephone interview. Data obtained from the maternal records included maternal history, prenatal screening tests, imaging results from anatomical ultrasound and fbMRI scans, maternal CMV status, and perinatal history. Data retrieved from children’s medical records included gestational week, birth weight, and head circumference; results of complete blood count, serum liver enzymes, head ultrasound, retinal examination, and hearing evaluation; and reports from the pediatric infectious diseases clinic follow-up visits including information regarding antiviral treatment, where children were seen at regular intervals during their first two years of life.

### 2.2. MRI Scans

Fetal brain MRI was performed using a 1.5-Tesla system (Optima scanner, GE Healthcare Technologies, Milwaukee, WI, USA). Single-shot fast spin echo T2-weighted sequences in three orthogonal planes were performed using the half Fourier technique (NEX = 0.53) with the following parameters: section thickness of 3 or 4 mm, no gap, flexible coil (8-channel cardiac coil), field of view (FOV) was determined by the size of the fetal head with a range of 24 × 24 cm to 30 × 30 cm, acquisition time between 40 and 45 s, matrix 320/224, TE 90 ms, TR 1298 ms, pixel bandwidth 122 Hz/pixel, and SAR values were between 1.1 and 1.7 W/kg. All MRI scans were performed by the same protocol, carried out in our institution, and assessed by a specialist in fetal US and fbMRI (EK) and a neuro-radiologist MRI expert (CH), as previously published [21].

### 2.3. White Matter T2 Hyper-Intense Signal (WMHS)

WMHSs, depicted in Figure 1, are a qualitative, subjective finding from the fetal brain white matter on T2 sequences. Fast T2-weighted sequences remain the basis of fetal MRI and are the most frequently used. As such, a consistent protocol was followed: two specialists, EK and CH, with twenty plus years of experience using the same protocol which considers gestational age, interpreted the fbMRI separately. 

In this way, WMMSs were assessed in the subplate zone by each lobe separately. 

### 2.4. Interobserver Validity of Measurements

Only cases of complete agreement on the presence or absence of WMHSs were included in the study [6]. In cases of disagreement among the assessors on the localization of WMHSs, only the undisputed interpretation was included in the study. Accordingly, fetuses were allocated to a WMHS positive and negative group.

### 2.5. Neurodevelopmental Assessment

The follow up consisted of a formal assessment of children using the Vineland Adaptive Behavior Scales (VABS) for cognitive and motor evaluation at the Sheba Medical Center Child Development Center.

VABS is a structured parent interview assessing four different domains of behavior: communication, daily living skills, socialization, and motor skills. All four domains are included in an adaptive score composite [15]. VABS interviews were conducted by two medical students trained and supervised by a pediatric neurologist and child development expert experienced in conducting VABS. Inter-rater reliability was assessed by comparison of VABS scores in 15 cases that were evaluated by the pediatric neurologist and the medical students. This validation demonstrated high concordance in VABS scores (Pearson’s correlation coefficient 0.88, B = 1.15).

The assessment was performed at one point in time. 

### 2.6. Hearing Evaluation

The follow-up consisted of a hearing evaluation at the Sheba Medical Hearing & Speech Center.

Infants were tested using transient evoked otoacoustic emissions as part of a universal hearing screening test performed on all infants prior to hospital discharge and a comprehensive audiological evaluation including diagnostic auditory brainstem responses (ABRs) testing to clicks and tonal stimuli presented by air conduction and, when necessary, by bone conduction, behavioral audiometry, tympanometry, and/or otoscopic examination by an ear, nose, and throat specialist. Audiologic evaluation was performed within 10–14 days after discharge. Thresholds to clicks and 1 kHz tone burst stimuli, i.e., ≤20 dB normal hearing level in both ears, were considered within the normal range. Periodic audiological follow-ups were also carried out in accordance with the recommendations of the Joint Committee on Infant Hearing 2007 Position Statement [16]. Hearing impairment was defined as unilateral or bilateral sensorineural hearing loss greater than 20 dB HL in the 500–4000 Hz frequency region based on elevated ABR thresholds for air and bone conduction for 1 and 4 kHz tone burst stimuli. Hearing impairment severity was determined based on Clark’s classification [17].

Categorical variables were expressed as numbers and percentages. The distribution of continuous variables was assessed using histograms and Q-Q plots. Continuous variables were described using median and interquartile range (IQR) or mean and standard deviation (SD). Categorical variables were compared using the Chi-square test or Fisher’s exact test as appropriate. Continuous variables were compared using Student’s *t*-test or the Mann–Whitney test, as appropriate. Multivariate linear regression was used to evaluate the association between the trimester (1st/2nd), WMHSs, and valganciclovir treatment and each of the Vineland scores. The linear regression was evaluated to meet the assumptions. Patients were matched according to infection trimester and compared using the McNemar test (categorical variables) or Wilcoxon test (continuous variables).

A two-tailed *p* < 0.05 was considered statistically significant. Analyses were performed with SPSS (IBM SPSS Statistics for Windows, Version 24.0, 2016. IBM Corp., Armonk, NY, USA).

### 2.7. Ethics Approval

The study was approved by the Sheba Medical Center institutional review board. Informed consent was requested via letters sent to eligible cases prior to telephone calls and approved in the VABS interview.

## 3. Results

### 3.1. Maternal and Pregnancy Demographic, Clinical, and Imaging Characteristics

The study subjects comprised 58 singleton pregnant women that underwent fbMRI scans (Table 1). All women were in their third trimester of pregnancy at the time of the fbMRI scan, with a median gestational age of 32 weeks (IQR 32–34). The average maternal age was 31.5 years (SD 4.1).

The cohort was divided into 21 fetuses with WMHS findings and 37 fetuses with a normal fbMRI, as defined in Section 2.4. There was no significant difference in the epidemiological and clinical features between the two groups except for the suspected week of the maternal CMV infection. The median timing of maternal CMV infection in the WMHS group was detected in the first trimester at 8 weeks (IQR 6.5–19.5), whereas for the normal MRI group it was in the second trimester at 17 weeks (IQR 11–23, *p* = 0.015).

In the WMHS group, the hyper-intense signals were depicted in the temporal lobes in all fetuses, and in 62% of the fetuses, it was depicted diffusely including the frontal and parietal lobes.

### 3.2. Postnatal Clinical and Imaging Characteristics

Delivery characteristics, hearing, and head ultrasound assessments after birth are described in Table 2. The two groups were not significantly different in birth weight and head circumference. The Polymerase Chain Reaction (PCR) for CMV was positive in saliva and urine for all infants. None of the neonates in the cohort had thrombocytopenia, hepato-splenomegaly, petechiae, or chorioretinitis. The presence of lenticulostriate vasculopathy (LSV) was not significantly different between the groups (*p* = 0.426). No difference was found in hearing assessment after birth by transient evoked optoacoustic emission and by auditory brainstem response (*p* > 0.999). There was a higher proportion of neonates with subependymal pseudocysts (SEPC) in the head ultrasound in the WMHS group (*p* = 0.043). Interestingly, the WMHS group had a significantly higher portion of children treated by valganciclovir, 67%, in comparison to 11% in the normal MRI group (*p* < 0.001). We speculate that this was because clinicians were more inclined to intervene if there was already evidence of white matter lesions or SEPC.

### 3.3. Long-Term Neurodevelopmental and Hearing Outcome

A description of the functional outcomes as measured by the VABS score and hearing assessments is presented in Table 3. VABS was assessed at a younger median age, 2.3 years (IQR 1.6–3.5), in the normal fbMRI group in comparison to 3.4 years (IQR 2.4–4.5) in the WMHS group (*p* = 0.03). In both groups, the median scores were within the normal range. There was no difference between the groups in the four developmental domains of VABS and in the adaptive composite score.

The week of maternal infection was earlier in the WMHS group and earlier CMV infection during pregnancy is associated with an increased likelihood for an abnormal developmental outcome [18]. In order to overcome this difference, the group comparison was matched by week of infection. After matching, each group consisted of eighteen children. All VABS scores in all four domains and in the adaptive composite score were within the normal range in both groups. The only significant difference was in the median score of the motor skills, which was higher in the WMHS group than the normal MRI group (106, IQR 97–116.3 and 100.5, IQR 77.5–105.3, respectively, *p* < 0.032). After matching, the ratio of children treated with valganciclovir was higher in the WMHS group (11/18 vs. 1/18, *p* = 0.002). 

Testing the effect of antiviral treatment and WMHSs on outcome was conducted by multivariate regression. Four independent variables were used: week of infection and three variables of possible situations of the presence of WMHS and valganciclovir treatment. The three situation conditions were: (1) WMHS +/valganciclovir −, (2) WMHS −/valganciclovir +, and (3) WMHS +/valganciclovir +. These situations were compared to the reference situation which was WMHS −/valganciclovir −. Five multivariate regression models were constructed for five outcome measures including the four VABS domains and their composite score.

Comparing the effect of the different situations to the reference situation was insignificant except for the following case (Table 4): children with WMHSs in their fetal MRI that were not treated with valganciclovir (WMHS +/valganciclovir −) showed worse VABS motor outcome (slope = −22.2, *p* = 0.001) and VABS composite score (slope = −7.6, *p* = 0.051). Children with WMHSs in their fetal rMRI that were treated with valganciclovir (WMHS +/valganciclovir +) had better VABS social scores (slope = 2.5, *p* = 0.03).

## 4. Discussion

The aim of this study was to determine whether subtle imaging abnormalities expressed as WMHSs in fbMRI are associated with subsequent neurodevelopmental deficits by using VABS assessment and hearing outcomes, in children with congenital CMV infection.

White matter abnormalities are one of the most characteristic findings of congenital CMV, occurring in up to 75% of clinically symptomatic and 30% of asymptomatic newborns [6]. These hyperintensities may represent different lesions or normal stages of development. Therefore, a finding of WMHSs in fbMRI poses a dilemma both from a diagnostic and prognostic perspective. 

A previous study from our group [13] demonstrated that when WMHS was positively identified in fbMRI undertaken in fetuses with congenital CMV, apparent diffusion coefficient values were increased. This is consistent with the fact that WMHS in fetuses with congenital CMV is usually interpreted as interstitial edema and a potential sign for brain inflammation [12,15,22]. This finding has been corroborated in a subsequent retrospective 2023 study which concluded that neonatal white matter apparent diffusion coefficient values were significantly higher in patients with clinical impairments [6].

Before addressing the study question in detail, there were two important observations that emerged from the results. First, this study showed that fetuses expressing a positive WMHS on fbMRI had an earlier week of infection in the first trimester of pregnancy. This is a significant finding as the likelihood for abnormal development is higher in early CMV infection [23,24,25]. While severe cases may present with typical CMV imaging findings such as calcifications and microcephaly, in mild cases without noticeable developmental deficits, the sole presentation of early CMV infection could be WMHS. 

Secondly, another difference between the groups was the higher proportion of subependymal pseudocysts (SEPCs) in the WMHS group. SEPC is a common head ultrasound finding occurring in 0.5–5% of healthy term infants. SEPCs are mainly a result of the expected lysis of the germinal matrix [26,27]. The follow-up of fetuses with SEPC showed they had an overall good neurodevelopmental outcome. Poor outcome was associated with underlying etiology rather than the occurrence of SEPC [28,29,30]. No study, hitherto, tested the association of SEPC in children with congenital CMV and neurodevelopmental outcome.

Since WMHSs are an expression of an increase in fluid content in brain tissue, edema, and because of the established link with CMV infection, a process that evolves over a few weeks, only infections acquired in the first trimester will have enough time to fully develop [25]. Another explanation is based on the finding that CMV shedding in the amniotic fluid occurs roughly 7–8 weeks after maternal infection [25]. If one assumes that it would take a similar period of time for replication of the virus to have an effect on the brain, then the timing of the fbMRI at 32 weeks gestation may not reflect the full extent of the effect of the virus on the brain, as opposed to the first trimester infections, but a later scan may identify other radiological findings. 

Interestingly, although the WMHS group had a larger proportion of fetuses with a first trimester infection, the abnormal fbMRI signal detected was not subsequently correlated with neurodevelopmental outcome deficits—within the timeframe studied—nor with a hearing deficit. In fact, after matching the groups by trimester of infection, the WMHS group demonstrated better motor outcome findings on assessment. This finding suggests that in our sample, WMHSs are not a predictive factor for neurodevelopmental outcome deficits. Thus, the increased fluid content in brain tissue, reflected as WMHSs, is not necessarily part of a destructive process of brain tissue and may even potentially be part of a protective process. 

Moreover, the neurodevelopment of the children in both groups was within the normal range even after matching for week of CMV infection. The median VABS assessment time was earlier in the WMHS group, 28 months, compared to the normal MRI group, 40 months. This difference should not affect the comparison as the VABS is normalized for age. Moreover, the median age was older than 24 months in both groups. At this age, the development of children in all four domains is mature enough for developmental deficits to be recognized [28].

One factor which partially complicates the simplistic explanation is the higher proportion of neonates treated with valganciclovir in the WMHS group. Valganciclovir is an antiviral drug administered for congenital CMV with specific clinical signs and symptoms [5]. Only three patients, with hearing impairment on the auditory brain stem response after birth, satisfied these criteria to initiate treatment [11]. However, 4 (11%) patients in the normal MRI group and 14 (67%) in WMHS group were treated. The higher proportion of treated infants in this group reflects the concern of physicians facing abnormal neuroimaging findings even if the association of those findings with neurodevelopmental deficit is unclear. All the children in our study were treated for 6 months; thus, the treatment might have caused a bias of the outcome of the WMHS group. 

As treatment with valganciclovir may affect neurodevelopment, we undertook a multivariate analysis to test the effects of infection week and the four different situations with the presence of WMHSs and valganciclovir treatment on neurodevelopmental outcomes. The results showed that the timing of infection did not independently affect the outcome. Motor scores (and as a result, the composite score too) were negatively affected by a lack of treatment in the WMHS group. Kimberlin et al. demonstrated similar results: six months of valganciclovir treatment led to a modestly improved neurodevelopmental outcome in infants with moderate-to-severe CMV symptoms [11]. Our results raise the possibility that withholding antiviral treatment in children with WMHSs prevented them from fulfilling their motor development potential. The meaning of this finding is limited by the small size of our study. 

Although our finding is suggestive of a positive effect of antiviral treatment on motor performance in cases of congenital CMV infection and WMHSs on fbMRI, the decision to start antiviral treatment should be made with caution and consider the possible adverse events of valganciclovir of neutropenia or a theoretic toxic effect on gonads [11,31]. As fbMRI is becoming more common, larger studies could assess the significance of minor findings such as WMHSs and provide information regarding the advantage of antiviral treatment in these cases.

In terms of limitations, this is a historic prospective case study with a limited number of age-matched compared groups. We were not able to compare with healthy controls, only with fetuses with CMV; therefore, there is a potential inherent bias. 

We performed the VABS assessments at a median age of just over 24 months in both groups. Follow-up was not performed beyond this age when other development issues may emerge. The median VABS assessment age was lower in the WMHS group, 28 months, compared to the normal MRI group, 40 months. Although we normalized for age, this could still cause a difference especially with finer motor skills and concentration.

Lastly, the confounder effect of valganciclovir prevents the potential to more firmly suggest that WMHSs are not associated with an abnormal neurodevelopmental outcome.

The strength of the study is the detailed descriptions of the cases and the ability to track long-term detailed developmental assessments.

## 5. Conclusions

In our limited study groups, we failed to find an association between WMHSs and neurodevelopmental outcomes or hearing deficit. We observed that treatment with valganciclovir is associated with a better neurodevelopmental outcome in the WMHS group. 

Larger studies should be performed to assess the effect of antiviral treatment on future neurodevelopmental and hearing outcomes of infants with congenital CMV following minor imaging signs such as WMHSs. 

## Figures and Tables

**Figure 1 diagnostics-14-00797-f001:**
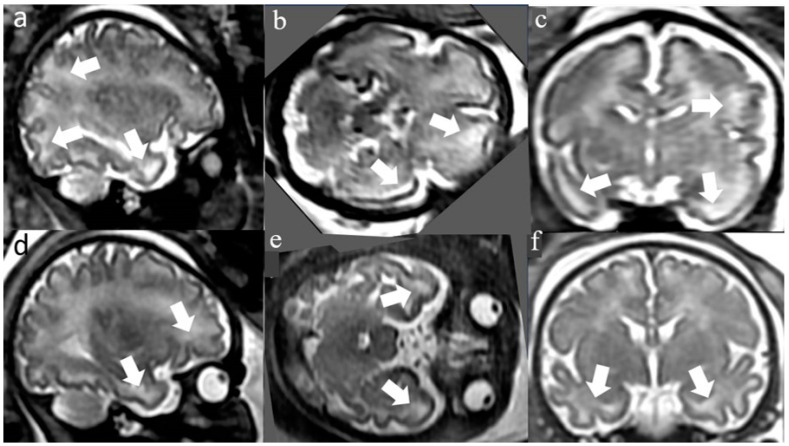
White matter hyperintensities *. * T2 axial and coronal brain MR images (single-shot fast spin echo T2-weighted sequences in three orthogonal planes using half Fourier technique, NEX = 0.53) at 32 weeks gestational age. The first fetus (**a**–**c**) and second fetus (**d**–**f**) with diffuse WMHSs. The WHMSs are marked by white arrows.

**Table 1 diagnostics-14-00797-t001:** Demographic and clinical characteristics ^.

Characteristic	Normal MRI	WMHS	*p*-Value
Maternal age, years, mean (SD)	32.5 (4.1)	30.9 (4.7)	0.191
Gestational age at MRI, median (IQR)	34 (32–35)	33 (33–34)	0.593
Pregnancy number, median (IQR)	2 (2–3)	2 (2–3)	0.670
Labor number, median (IQR)	1 (1–2)	1 (1–1.75)	0.652
Abnormal outcome in previous labors, n (%)	0 (0)	3 (15)	0.052
Abnormal maternal medical background, n (%)	2 (6.1)	1 (4.8)	>0.999
Spontaneous conception, n (%)	28 (82.4)	21 (100)	0.072
Gender (female), n (%)	13 (35.1)	11 (52.4)	0.200
Abnormal nuchal translucency scan, n (%)	1 (3.1)	1 (5.3)	>0.999
Abnormal 1st trimester biochemical test, n (%)	1 (3.7)	0 (0.0)	>0.999
Abnormal 2nd trimester biochemical test, n (%)	1 (3.8)	0 (0.0)	>0.999
Abnormal early anatomical scan, n (%)	0 (0)	0 (0)	>0.999
Abnormal late anatomical scan, n (%)	3 (8.1)	1 (4.7)	>0.999
Infection week, median (IQR)	17 (11–23)	8 (6.5–19.5)	0.015

^ MRI—magnetic resonance imaging; WMHS—white matter hyper-intense signal; IQR—interquartile range.

**Table 2 diagnostics-14-00797-t002:** Postnatal short-term outcome measures.

Clinical and Radiological Findings	Normal MRI (N = 37)	WMHS (N = 21)	*p*-Value
Duration of pregnancy (weeks), median (IQR)	39 (37.6–39.9)	39.3 (38.4–39.9)	0.633
Birth weight, (gr) median (IQR)	3114 (2878–3410)	3052 (2845–3420)	0.974
Birth weight (percentile), median (IQR)	52 (32–73)	41 (29.5–73.5)	0.840
Head circumference (cm), median (IQR)	34 (33–35)	34 (33–35)	0.647
Head circumference (percentile), median (IQR)	51 (12–79)	54 (29–80)	0.510
Head circumference < 10%, n (%)	1 (11%)	5(14.3%)	>0.999
SEPC in head ultrasound. n (%)	1 (3%)	4 (21%)	0.043
LSV in head ultrasound, n (%)	4 (11%)	4 (21%)	0.426
Any abnormal finding in head US, n (%)	5 (14%)	6 (29%)	0.296
Abnormal acoustic emissions, n (%)	1 (3%)	1 (5%)	>0.999
Abnormal auditory brain response (after birth), n (%)	2 (5%)	1 (4.8%)	>0.999
Valganciclovir treatment, n (%)	4 (11%)	14 (67%)	<0.001

SEPC—subependymal pseudocysts; LSV—lenticulostriate vasculopathy.

**Table 3 diagnostics-14-00797-t003:** Comparison of VABS scores matched for week of infection.

Clinical and Radiological Findings	Normal MRI (N = 37)	WMHS (N = 21)	*p*-Value
Age at VABS, (years), median (IQR)	2.3 (1.5–3.5)	3.8 (2.5–4.5)	0.049
VABS motor skills, median (IQR)	100.5 (77.5–105.3)	106 (97–116.3)	0.032
VABS daily living skills, median (IQR)	109 (99.25–115.3)	105 (97.5–116.8)	0.641
VABS socialization skills, median (IQR)	106 (100.3–114)	107 (97.3–116)	0.501
VABS communication skills, median (IQR)	103 (100.8–107.3)	102 (94.3–108.8)	0.469
VABS adaptive score composite, median (IQR)	102.5 (94–111.3)	106 (98.3–110)	0.233
Hearing impairment; n (%)	2 (11.1)	3 (16.7)	>0.999

**Table 4 diagnostics-14-00797-t004:** Multivariate regression models of developmental outcome.

Multivariate Regression	Motor Standard Score		Social Standard Score		Daily Skills Standard Score		Communication Standard Score		General Standard Score	
	Slope * (CI)	*p*-Value	Slope * (CI)	*p*-Value	Slope * (CI)	*p*-Value	Slope * (CI)	*p*-Value	Slope * (CI)	*p*-Value
Constant	103		99		105		105		103	
Infection week (week)	4.8 (−1.7–4.1)	0.26	−2.6 (−4.6–4.1)	0.91	−1.0 (−7.8–5.8)	0.76	2.7 (−3.7–9.2)	0.39	1.8 (−1.7–7.5)	0.51
WMHS+/ Valganciclovir−	−22.2 (−34.7–−11.3)	0.001	2.1 (−2.0–8.1)	0.49	−3.6 (−13.0–5.7)	0.44	−0.8 (−9.7–8.1)	0.85	−7.6 (−15.3–0.1)	0.051
WMHS−/ Valganciclovir+	−4.2 (−19.0–10.5)	0.56	3.7 (−5.1–10.2)	0.51	−9.1 (−20.1–2.7)	0.13	−9.8 (−21.0–1.4)	0.08	−6.5 (−16.2–3.2)	0.18
WMHS+/ Valganciclovir+	0.4 (−9.5–10.5)	0.93	2.5 (0.5–10.8)	0.03	−3.8 (−4.1–11.5)	0.34	2.9 (−4.6–10.5)	0.45	3.9 (−2.6–10.5)	0.24

* Average increase in outcome score per unit increase of predictor; CI 95%, Confidence Interval; WMHS, white matter hyper-intense signal.

## Data Availability

No new data were created or analyzed in this study. Data sharing is not applicable to this article.

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
