# Peer review of "Association between White Matter T2 Hyper-Intense Signals in Fetal Brain Magnetic Resonance Imaging and Neurodevelopment of Fetuses with Cytomegalovirus Infection"

_diagnostics, 2024, doi:10.3390/diagnostics14080797_

Round 1

Reviewer 1 Report

Comments and Suggestions for Authors

Prenatal CMV infection is an important health problem. This article may contribute to the literature. After revision, the article should be re-evaluated.

1. While the study is stated to be prospective in the material and method section, it is stated in the limitations that it is retrospective. What is the reason for this contradiction?

2. The cases belong to the years 2011-2015. Why was the article written after so long?

3. The groups should be expressed more clearly in the Abstract section.

4. There is no relationship between the WMHS finding on MRI and the outcome of the cases. This result should be discussed in more detail.

Author Response

Reviewer 1

  1. While the study is stated to be prospective in the material and method section, it is stated in the limitations that it is retrospective. What is the reason for this contradiction?

Thank you for pointing out – yes this is an historic prospective case study. The mistake in the Methods section was corrected accordingly.

  1. The cases belong to the years 2011-2015. Why was the article written after so long?

We originally hoped to publish in 2020 however because of COVID and its aftermath it was it difficult to galvanize the research group and during a recent review, we decided to complete the study and submit.

  1. The groups should be expressed more clearly in the Abstract section.

Thank you for pointing out. We corrected and added an additional sentence so that we hope that it is entirely clear now.

  1. There is no relationship between the WMHS finding on MRI and the outcome of the cases. This result should be discussed in more detail.

Thank you for the comment.

Since many of the children with the WMHS were treated with Valacyclovir, we tested the effect of anti viral treatment on the outcome by muiltivariate regression (see lines 241-254) and there we showed that children with the signal that were not treated had worse outcomes compared to those who were treated with Valacyclovir had better outcomes.

In lines 324-341 (new document), in the discussion section, we discuss in depth the possible effect of the antiviral treatment as a confounder (also referenced in lines 351-2 of the limitations section).

Reviewer 2 Report

Comments and Suggestions for Authors

Dear Authors,

Thank you for making this work available.

Your retrospective study compares outcomes between 2 different CMV infected fetuses: those that present T2 white matter hyper-intense signals (WMHS) and those without, at a routine fetal brain MRI around 32-24 weeks of gestation. You have indeed interesting results. One would expect that WMHS would be a predictor of poor developmental outcome, but this, as you recognize in your Discussion, could have been mitigated by the higher use of antiviral medication. An important limitation is indeed, as you recognize, lacking the control group - healthy fetuses at 32-34 weeks that undertake fbMRI. Nevertheless, I think your work should be published as it adds up to our understanding of what happens in fetal CMV infections and what clinicians should be aware of.     

I have some questions that I would kindly ask you to clarify and some minor comments:

1. Why is your cohort so "old" - why did you include only fbMRI undertaken between 2011 to 2015? Why not expend your cohort to more recent years?

2. CMV 77 DNA in neonatal saliva, confirmed by a urine specimen during the first 2 weeks of life - please clarify

3. White Matter T2 Hyperintense Signal is a subjective qualitative assessment - is this routine practice? Has this been described by others?

4. Figure 1 - the text has jumped and is a paragraph up before the Figure itself

5. The cohort was prospectively followed, however, your study is retrospective in nature - you looked back at the data from 2011-2015?

6. Since WMHS is an expression of an increase of fluid content in brain tissue, edema, 289 and because of the established link with CMV infection, a process that evolves over a few weeks, only first trimester infections will have enough time to fully develop - acquired in the first...   

7. For the Abstract - please rectify that it was a retrospective analysis

Thank you for your work.

Author Response

  1. Why is your cohort so “old”, why did you include only fbMRI undertaken between 2011 to 2015? Why not expend your cohort to more recent years?

We originally hoped to publish in 2020 however because of COVID and its aftermath it was it difficult to galvanize the research group and during a recent review, we decided to complete the study and submit.

  1. CMV 77 DNA in neonatal saliva, confirmed by a urine specimen during the first 2 weeks of life - please clarify.

Thank you for your comment. The saliva test is highly sensitive but has false positive results therefore we use as a screening test. The gold standard, more specific test, is the urine PCR. Therefore, in order to include all true positives, we confirmed with the sensitive urine test.  We only included cases where urine PCR confirmed the diagnosis.

We added to methods section line 77, a clarification so that it ties in.

  1. White Matter T2 Hyperintense Signal is a subjective qualitative assessment - is this routine practice? Has this been described by others? ??? Aren't we describing results about WHMS

This has become routine in our group because of the previous research performed by members of our group and others that have shown there is potential diagnostic value of this signal and it is something that we have been accumulating data on.

  1. Figure 1 - the text has jumped and is a paragraph up before the Figure itself.

Corrected

  1. The cohort was prospectively followed, however, your study is retrospective in nature – you looked back at the data from 2011-2015?

This was a historic prospective study. We retrieved the data retrospectively.

  1. Since WMHS is an expression of an increase of fluid content in brain tissue, edema, 289 and because of the established link with CMV infection, a process that evolves over a few weeks, only first trimester infections will have enough time to fully develop.

Corrected

  1. For the Abstract - please rectify that it was a retrospective analysis

Corrected

Round 2

Reviewer 1 Report

Comments and Suggestions for Authors

The authors made the necessary corrections in line with the suggestions. I think the final version of the manuscript can be published in the journal.